# Acquired Resistance to Decitabine Associated with the Deoxycytidine Kinase A180P Mutation: Implications for the Order of Hypomethylating Agents in Myeloid Malignancies Treatment

**DOI:** 10.3390/ijms26115083

**Published:** 2025-05-25

**Authors:** Kristina Simonicova, Lubos Janotka, Helena Kavcova, Ivana Borovska, Zdena Sulova, Albert Breier, Lucia Messingerova

**Affiliations:** 1Institute of Molecular Physiology and Genetics, Centre of Biosciences, Slovak Academy of Sciences, Dúbravská cesta 9, 840 05 Bratislava, Slovakia; kristina.simonicova@savba.sk (K.S.); lubos.janotka@upol.cz (L.J.); helena.kavcova@savba.sk (H.K.); ivana.sevcikova@savba.sk (I.B.); zdena.sulova@savba.sk (Z.S.); 2Department of Biology, Faculty of Medicine and Dentistry, Palacký University Olomouc, Hnevotinska 3, 77515 Olomouc, Czech Republic; 3Institute of Biochemistry and Microbiology, Faculty of Chemical and Food Technology, Slovak University of Technology in Bratislava, Radlinského 9, 812 37 Bratislava, Slovakia

**Keywords:** acute myeloid leukemia, DAC, deoxycytidine kinase, hypomethylating agents, resistance, pyrimidine synthesis

## Abstract

The backbone of therapy for elderly patients with myelodysplastic syndromes and acute myeloid leukemia consists of hypomethylating agents 5-aza-2’-deoxycytidine (DAC) and 5-azacytidine (AZA). However, resistance frequently emerges during treatment. To investigate the mechanisms of resistance, we generated DAC-resistant variants of the acute myeloid leukemia cell lines, MOLM-13 and SKM-1, through their prolonged cultivation in increasing concentrations of DAC. The resistant cell variants, MOLM-13/DAC and SKM-1/DAC, exhibited cross-resistance to cytarabine and gemcitabine, but remained sensitive to AZA. Existing studies have suggested that the loss of deoxycytidine kinase (DCK) may play an important role in DAC resistance. DCK is critical for DAC activation, but the precise mechanisms of its downregulation remain incompletely understood. We identified a novel point mutation (A180P) in *DCK*, which results in acquired DAC resistance. Although the *DCK* mRNA was actively transcribed, the mutant protein was not detected in DAC-resistant cells. The transfection of HEK293 cells with the mutant *DCK,* combined with proteasomal inhibition, revealed rapid proteasomal degradation, establishing a mechanistic link between the A180P mutation and DCK loss, not previously described. This highlights the importance of also evaluating DCK at the protein and/or enzymatic activity levels in patients. The loss of functional DCK impairs the phosphorylation of deoxynucleosides, conferring resistance to DAC, gemcitabine, and cytarabine, but AZA, phosphorylated by uridine–cytidine kinase, remains effective and may represent a therapeutic alternative for patients with acquired DAC resistance.

## 1. Introduction

Myelodysplastic syndromes (MDSs), recently re-termed myelodysplastic neoplasms in the 5th edition of the WHO classification [1], and acute myeloid leukemia (AML) are heterogeneous groups of hematologic malignancies that primarily affect elderly patients. The only curative therapy for these diseases is hematopoietic stem cell transplantation. However, many elderly patients are ineligible for such treatment, due to various comorbidities. In these patients, low-intensity treatment is used with two hypomethylating agents (HMAs), 5-aza-2′-deoxycytidine (DAC) and 5-azacytidine (AZA), playing a primary role [2,3]. After stepwise phosphorylation by pyrimidine nucleotide kinases, AZA is converted into AZA triphosphate and incorporated into RNA. Only about 15% of AZA diphosphate is reduced to DAC diphosphate, which, following further phosphorylation, can be incorporated into DNA. In contrast, DAC undergoes stepwise phosphorylation and is incorporated exclusively into DNA. When DAC is incorporated into DNA, it prevents DNA from becoming methylated and inhibits DNA methyltransferases [4].

Typically, DAC and AZA are administered to patients for 5 or 7 consecutive days in 28-day cycles. Although this treatment is not curative, it can reduce transfusion dependency, improve health-related quality of life, delay progression from MDS to AML, and prolong overall survival in patients who are ineligible for intensive therapy. However, at least four to six cycles of therapy are required to achieve a response to HMAs, and, if tolerated, treatment is continued until disease progression [5,6,7].

Unfortunately, with such long-term treatment, the development of resistance to HMAs is almost inevitable. Resistance to HMAs is still not fully understood; however, several mechanisms have been proposed, including impaired drug transport, enhanced drug metabolism, and the loss or dysfunction of activating enzymes, such as deoxycytidine kinase (DCK, reviewed in [8]). Moreover, the subsequent treatment of patients with relapsed/refractory high-risk MDS remains limited [9]. Therefore, it is necessary to elucidate the molecular basis of HMA resistance in order to determine a strategy to overcome it.

In a previous paper, we described three variants of AZA-resistant cells generated from human AML cell lines, MOLM-13 and SKM-1, by passaging cells with progressively increasing concentrations of AZA. We observed that these resistant cell variants differ from each other, but alternations in the pyrimidine salvage pathway are common to all of them [10]. In addition to resistance itself, perhaps an even greater problem in the treatment of hematological malignancies is the cross-resistance or multidrug resistance to other drugs to which the cells have not been exposed [8]. An example is the cross-resistance to DAC and the other deoxycytidine antimetabolites, cytarabine (AraC) and gemcitabine (GEM), observed in some AZA-resistant variants of the cells described above [10]. This phenomenon limits the treatment options available and must be taken into account when switching therapies.

In the present paper, we focus on acquired DAC resistance, studying the DAC-resistant cell variants of SKM-1 and MOLM-13 cells generated in our laboratory. Both cell lines, SKM-1 and MOLM-13, are established from the peripheral blood of patients with AML, following an initial MDS diagnosis. 

## 2. Results

### 2.1. Resistance to Deoxycytidine Analogs

Two DAC-resistant cell variants were prepared in our laboratory from the SKM-1 cell variant (SKM-1/DAC—S/D) and the MOLM-13 cell variant (MOLM-13/DAC—M/D). The MOLM-13/DAC subline has already been described in our previous paper [11]. Using an MTS assay, we determined the IC_50_ values of four (deoxy)cytidine analogs, DAC, AZA, GEM, and AraC, in our cell variants (Figure 1A). Neither of our two DAC-resistant variants show cross-resistance to AZA; however, both variants are extensively cross-resistant to GEM and AraC. In the case of DAC and AraC, the IC_50_ is higher than 40 μM. At this concentration, there is still no detectable effect on cell viability (Figure 1B,E). The IC_50_ for GEM is around 8 μM in both DAC-resistant cell variants.

### 2.2. Alterations in DNA Methylation and DNA Damage Induced by HMAs

For DAC’s mechanism of action to occur, its incorporation into DNA is essential. During maintenance methylation, DNA methyltransferases 1 (DNMT1) form a covalent intermediate with a carbon at position 6 of the cytidine targeted for methylation, by binding to the SH group of an essential cysteine at the enzyme’s active site [12]. Upon methylation of the cytidine carbon at position 5, this intermediate is cleaved. However, if a nitrogen atom is present instead of carbon at position 5, methylation does not occur, and the DNMT1 remains bound to the DNA strand, which disrupts the methylation of the daughter DNA [13].

To determine whether DAC and AZA induce changes in DNA methylation in our cell variants, we assessed the level of DNMT1 (Figure 2A) and the global DNA methylation status (Figure 2B) in cells treated with these drugs. We observed decreased DNMT1 protein levels and reduced global DNA methylation in all the cell variants treated with AZA. The treatment of parental SKM-1 or MOLM-13 cells with DAC caused a decrease in the DNMT1 levels and global methylation. In contrast, treatment of the DAC-resistant SKM-1/DAC and MOLM-13/DAC variants with DAC resulted in unchanged, or even increased, DNMT1 protein levels compared to the untreated control, with only insignificant changes in global DNA methylation. Comparing the parental and DAC-resistant variants, we can see that the long-term cultivation of parental cell lines with DAC led to decreased global DNA methylation in the SKM-1/DAC cell variant, but not in MOLM-13/DAC (Figure 2B).

In addition to DNA demethylation, HMAs can also induce DNA damage [14]. To determine whether DAC and AZA induce such effect in our cell variants, we assessed the phosphorylation of histone H2AX (γ-H2AX) in the cells treated with these drugs (Figure 3). The phosphorylation of histone H2AX at Ser139, an early cellular response to double-strand breaks, is a well-established marker of DNA damage. We decided to measure it at two different time points to capture both early and late responses to drug treatment. We observed increased H2AX phosphorylation in all the cell variants treated with AZA, as well as in SKM-1 or MOLM-13 cells treated with DAC. In DAC-resistant SKM-1/DAC and MOLM-13/DAC variants, treatment with DAC did not induced changes in the γ-H2AX levels.

Not surprisingly, AZA exerted multiple effects on DAC-resistant SKM-1/DAC and MOLM-13/DAC variants, whereas DAC did not, while both HMAs exerted effects on the parental cells. The presence of a decrease in DNMT1, the total methylation levels, and an increase in histone H2AX phosphorylation, suggested that HMA incorporation into DNA was occurring in all cases except for the DAC treatment of DAC-resistant cells. The decrease in DNMT1 levels in SKM-1 and MOLM-13 cells treated with AZA and DAC and in SKM-1/DAC and MOLM-13/DAC cells treated with AZA suggests that DNMT1 remains covalently bound to the total DNA fraction or has already been degraded by the proteosome.

### 2.3. Expression of Genes Related to Metabolism and Transport of the HMAs

For HMA to be incorporated into DNA, it must enter the cell and undergo activation through phosphorylation. We, therefore, examined the expression of genes encoding proteins responsible for the following processes: (i) nucleoside transport, including *SLC29A1* and *SLC29A2*, which encode equilibrative nucleoside transporters 1 and 2, and *SLC28A1* and *SLC28A3*, which encode concentrative nucleoside transporters of the SLC28 family; (ii) HMA activation, including *DCK* (encoding deoxycytidine kinase, which phosphorylates DAC), *UCK1*, and *UCK2* (uridine–cytidine kinases 1 and 2, which phosphorylate AZA), and *RRM1*, *RRM2*, and *RRM2B* (subunits of ribonucleotide reductase required for the conversion of AZA to its deoxy form for DNA incorporation); (iii) HMA deactivation, including *NT5C3A* (encoding 5′-nucleotidase) and *CDA* (encoding cytidine deaminase); and (iv) de novo pyrimidine synthesis, including *DHODH* (encoding dihydroorotate dehydrogenase), a key enzyme in this pathway. At the mRNA level (Figure 4A), we observed a 40–50% downregulation of *DCK* in both DAC-resistant variants. We also noted an upregulation of *NT5C3A* in MOLM-13/DAC and of *UCK2* in SKM-1/DAC (although *UCK2* upregulation was not statistically significant in MOLM-13/DAC). The upregulation of *DHODH* was detected in both DAC-resistant variants. *CDA*, *SLC28A1*, and *SLC28A3* were not expressed in any of our cell lines. Additionally, we examined the cellular levels of DCK, UCK1, and UCK2 proteins, as these are the initiating enzymes in HMA phosphorylation. Although *DCK* mRNA expression was reduced but still detectable in the DAC-resistant variants, we did not observe a detectable amount of DCK protein. The changes in UCK1 and UCK2 protein levels were not significant, although there was a trend toward the downregulation of UCK1 and the upregulation of UCK2 (Figure 4B,C). 

### 2.4. New A180P Mutation of DCK Associated with DAC Resistance

To elucidate the reason why the DCK protein is not found in SKM-1/DAC or MOLM-13/DAC cells despite the presence of mRNA produced by the transcription of this gene, we addressed this question in more detail. First, we sequenced the RT-PCR product of the entire coding region of the *DCK* gene and found a point substitution of guanine for cytidine in both DAC-resistant variants (Figure 5A). This missense mutation results in a substitution of alanine for proline at position 180 of the DCK protein. As indicated in the crystal structure (PDB entry: 1P61; [15,16]), this mutation site is located in close proximity to the ATP-binding site (Figure 5B).

We used four in silico tools to predict the potential effect of this mutation on the protein (Table 1). While the MutationTaster 2021 [17] and PolyPhen-2 [18] predicted that the protein features might be affected by this mutation with high confidence, and SNPs&GO [19] with moderate confidence, the web tool, SIFT [20], predicted that the mutation would be tolerated (but with a score of 0.09, whereas a score of less than 0.05 is predicted to be deleterious). Therefore, we decided to study whether this mutation alone could have caused the protein to be undetectable in the cells. By inserting either the coding sequence of wild-type *DCK* (*DCK*_wt_) amplified from the parental MOLM-13 cell line or the coding sequence of mutant *DCK* (*DCK*_mut_) amplified from its resistant MOLM-13/DAC counterpart into the pcDNA3.1/myc-His vector, we prepared constructs expressing *DCK*_wt_ or *DCK*_mut_, respectively. After propagation in bacteria, the plasmids were extracted and validated using Sanger sequencing. The HEK293 cells were transfected with either a plasmid carrying *DCK*_wt_ or *DCK*_mut_, and the cells were harvested 24 and 48 h after transfection. The expression of the plasmid carrying *DCK*_wt_ and *DCK*_mut_ was determined by Western blotting, using an antibody against the myc tag. While we detected the tagged DCK protein after both incubation intervals when using *DCK*_wt_ from parental MOLM-13 cells, we did not detect the tagged DCK protein when using the *DCK*_mut_ from MOLM-13/DAC cells despite successful transfection with the plasmid (Figure 5C).

Then, 24 h after transfection with a DCK_mut_-containing plasmid, we added proteasome inhibitors, either MG132 at a concentration of 10 μM or bortezomib at a concentration of 20 nM, to the HEK293 cells and cultured them for another 24 h. We then monitored the levels of proteins derived from this plasmid, using an antibody against the myc tag. Both inhibitors, but especially bortezomib, induced a remarkably increased immunodetection of the myc tag compared to the cells in which the proteasome was not inhibited (Figure 5D). In the cells in which proteasomal degradation was not inhibited, we observed only weak immunoreactivity of the myc tag, despite the fact that we used a larger amount of plasmid in these experiments. These results clearly demonstrate that both transcription and translation occur from the mutant *DCK_mut_* gene and that the resulting mutant protein carrying the A180P mutation is degraded downstream in the proteasome.

We also sequenced the entire coding region of *UCK1* and *UCK2*, but we did not observe any mutations in these genes.

### 2.5. Effect of Teriflunomide, a Dihydroorotate Dehydrogenase Inhibitor

Given the absence of functional DCK, a key enzyme in the salvage pathway of pyrimidine deoxynucleotide synthesis, in our DAC-resistant variants, we hypothesized that these cells may rely more heavily on the de novo pathway of pyrimidine biosynthesis. Therefore, we determined the effect of the dihydroorotate dehydrogenase (DHODH) inhibitor (teriflunomide, TFN) on our cell variants. We examined its effect on the cell number, metabolic activity, and cell death (Figure 6). While the effect on MOLM-13/DAC was the same as on its parental cell line, in the case of SKM-1/DAC, we observed a slightly higher effect of the drug on cell metabolism and cell death compared to the SKM-1 cell line.

## 3. Discussion

In our previous paper, we described that two out of our three AZA-resistant cell variants, prepared through long-term cultivation with AZA, also showed reduced sensitivity to DAC [9]. Many other research groups have observed some degree of cross-resistance to DAC in their cell variants with secondary AZA resistance [21,22,23,24,25]. However, our DAC-resistant variants prepared through long-term cultivation with DAC do not show cross-resistance to AZA (Figure 1), similar to the majority of cell variants in other studies [24,25,26,27], even though some DAC-resistant cell variants that are cross-resistant to AZA have also been observed [23,25]. Nevertheless, clinical studies to date have mainly focused on the use of DAC in the treatment of patients after AZA-treatment failure and not the other way around. Consistent with the results obtained in regard to cell lines, this regimen does not seem to be optimal [28,29,30,31,32]. However, based on the studies on cell lines, the use of the reverse order of the drugs used in the treatment of MDS or AML seems to have greater potential. We are aware of only one study that tested AZA in patients after DAC-treatment failure, with a response rate of 40%. However, there were only 10 participants in the study [29]. We believe that further research should focus on testing this drug order in MDS and AML treatment.

In some European countries, including the Slovak and Czech Republic, clinicians cannot freely choose which HMA to use for a patient. Although both HMAs are approved for the treatment of MDS and AML, only AZA is currently reimbursed by health insurance bodies and DAC is only used in exceptional cases, according to national authorities. This also explains our inability to test our hypotheses on clinical samples.

In contrast to AZA sensitivity, both of our DAC-resistant variants show cross-resistance to AraC (Figure 1). This cross-resistance was also observed by Qin et al. (2009) [26]. In addition to HMAs, low doses of AraC (LDAC) can be used in MDS and AML patients who are ineligible for intensive chemotherapy. Cross-resistance between DAC and AraC could play a role in the decision process to use HMA after previous treatment with AraC, or vice versa, to use LDAC after treatment with HMAs. In addition to AraC, our DAC-resistant sublines also showed significant cross-resistance to GEM (Figure 1). This cross-resistance has also been observed in DAC-resistant colorectal cancer cells [27]. Although GEM is not used in the treatment of MDS and AML, it plays an important role in the treatment of several solid tumors [33]. In some of them, the use of DAC in combination with conventional drugs is currently being tested [34].

The rate-limiting step in (deoxy)cytidine analog activation is their phosphorylation into the corresponding monophosphate forms. This step is catalyzed by UCKs, in the case of AZA, and by DCK, in the case of DAC, AraC, and GEM. The downregulation of DCK in our two DAC-resistant cell variants (Figure 4), thus, explains the resistance to DAC, the cross-resistance to both AraC and GEM, and also the lack of cross-resistance to AZA. When DCK activity is reduced or absent in cells, deoxycytidine analogues (DAC, AraC or GEM) cannot be phosphorylated and, therefore, cannot be incorporated into DNA and exert therapeutic effects on cells [8,35]. We acknowledge as a limitation of this study that we did not perform re-expression experiments introducing wild-type DCK into the DAC-resistant MOLM-13/DAC and SKM-1/DAC cell variants to directly test whether DAC sensitivity would be restored. Although such experiments would provide additional functional validation, they are technically challenging, since AML cell line models are difficult to transfect efficiently, while maintaining high cell viability [36], and because repeated DAC administration over several days is required. On the other hand, these experiments would not provide new insights, as several studies have already demonstrated the positive correlation between DCK loss and DAC resistance [37,38,39,40]. Our work provides deeper insights into the molecular basis of this resistance. While mutations in the *DCK* gene have been reported [25,26,41], our work identifies a unique A180P mutation in both resistant variants, which leads to the transcription and translation of the mutant DCK mRNA, but the resulting protein is efficiently degraded by the proteasome, as demonstrated in HEK293 cells (Figure 5C, D). The accumulation of the mutated protein in our resistant variants could not be demonstrated directly, as these cells are highly sensitive to proteasome inhibition and undergo cell death, even at low concentrations of bortezomib [11], making it difficult to inhibit the proteasome effectively without inducing significant cytotoxicity. These findings not only expand the spectrum of known DCK mutations, but also provide mechanistic insight into how such mutations can contribute to therapeutic resistance through post-translational protein regulation and underscore the critical importance of assessing DCK at the protein level, rather than relying solely on mRNA expression, as post-transcriptional mechanisms, such as proteasomal degradation, can profoundly impact drug responsiveness, despite the presence of detectable mRNA transcripts. 

In DCK-deficient cells, the reduction in cytidine diphosphate to deoxycytidine diphosphate secures a pool of this nucleotide which, after further phosphorylation, is incorporated into the DNA, allowing replication to continue. If we use AZA at this point, it is phosphorylated in the same way as cytidine for RNA synthesis, and a proportion of it in the form of AZA diphosphate is reduced to DAC diphosphate and is suitable for incorporation into DNA after further phosphorylation. Therefore, in our DAC-resistant variants with the novel DCK mutation (A180P) described in this paper that demonstrate cross-resistance to AraC and GEM, treatment with AZA remains effective (Figure 1, Figure 2 and Figure 3). 

The results describing the mutation and downregulation of DCK may appear straightforward and unsurprising, as the downregulation of DCK has been observed previously in DAC-resistant cell lines [24,25,26,27]. However, the findings in clinical settings remain inconclusive. Notably, these studies measured DCK gene expression solely at the mRNA level [42,43,44]. In contrast, our two DAC-resistant lines demonstrated that the DCK protein might be absent, even when DCK gene transcripts are detectable. Moreover, a review of other studies highlights that changes in DCK expression at the protein level are not uncommon, even in cases where changes at the mRNA level are minimal or absent [25,26,27]. Additionally, we observed a similar phenomenon in our previously published AZA-resistant SKM-1 and MOLM-13 cell variants when analyzing the expression of UCK1 and UCK2, enzymes responsible for phosphorylating pyrimidine nucleotides into their respective monophosphates [10]. Thus, we propose that to assess the potential of DCK as a marker for DAC efficacy, its expression should also be evaluated at the protein level and/or its activity should be measured in malignant cells. Future clinical trials testing the efficacy of DAC should also evaluate the mutation status of DCK in patients both before and during treatment. To our knowledge, there are currently no published studies that have specifically investigated DCK mutations in AML patients who have relapsed after DAC treatment. One relevant study analyzed 16 MDS patients who relapsed following DAC therapy and did not detect DCK mutations. However, the clinical context of those cases is not fully described, including how relapse was defined, how many treatment cycles patients received prior to relapse, and which 16 of the 30 relapsed patients summarized in the table were selected for sequencing [42]. 

DAC resistance caused by the absence of DCK could be overcome by modifying the drug structure. Administration of DAC already in the monophosphate form would make it independent of DCK. However, the transport of charged compounds into the cells is problematic, so further modifications of the drug structure would be necessary. Following this idea, GEM has already been modified by scientific groups. Compounds have been prepared that are able to enter the cell independently of nucleoside transporters and, after enzymatic cleavage, are available in the cell in the form of monophosphate, without the need for DCK [45,46,47]. 

Due to the absence of the DCK protein in DAC-resistant cell variants, we expected them to be more sensitive to de novo pyrimidine synthesis inhibition than the parental cells. However, we observed only a small difference in the case of SKM-1/DAC and no difference in MOLM-13/DAC compared to SKM-1 and MOLM-13, respectively (Figure 6). It seems that the cells are not dependent on this pathway and probably rely mostly on the salvage pathway, using enzymes UCK1 or UCK2 to phosphorylate uridine and cytidine (Figure 7). This is also indicated by the high sensitivity of DAC-resistant variants to AZA (Figure 1). We are not aware of any studies testing DHODH inhibitors in DCK-deficient cells. However, the effect of 3-deazauridine (3-DU) on DCK-deficient cells has been studied in the past. 3-DU is an inhibitor of CTP synthetase, an enzyme responsible for the conversion of UTP to CPT. The use of 3-DU had a greater effect on DCK-deficient cells than on the parental cell line [48]. However, the inhibition of CTP synthetase affects not only the de novo pyrimidine synthesis pathway, but also the salvage pathway of pyrimidine synthesis from uridine, leaving DCK-deficient cells with the only option to synthesize dCTP from cytidine [10].

It appears that for AZA to function, the UCK2 enzyme is crucial, whereas UCK1 seems to preferentially phosphorylate naturally occurring cytidine and uridine. In our previous study, we demonstrated that TFN exhibits a synergistic effect with AZA in AZA-resistant cells with functional UCK2. However, this synergy was absent in cells harboring the UCK2 mutation, which is predicted to be deleterious [10]. Furthermore, it was shown that UCK2 exhibits considerably higher catalytic efficiency and affinity for cytidine and uridine compared to UCK1 [49]. In our DAC-resistant variants, we observed a trend toward the downregulation of UCK1 and the upregulation of UCK2 (Figure 4). We speculate whether this trend might represent an adaptative mechanism through which these cells compensate for DCK deficiency and the necessity for a higher rate of uridine and cytidine phosphorylation.

Taken together, our results reveal a novel DCK mutation (A180P) that causes proteasomal degradation of the mutant DCK protein. This mutation renders DAC, AraC, and GEM ineffective, while AZA remains active and retains its therapeutic effect in resistant cells. It should be emphasized that the loss of DCK expression cannot be reliably detected by qPCR alone; quantification of the DCK protein is necessary. Another important implication of our current results, when considered alongside our previous findings on AZA-resistant cell variants [10], is that administering DAC treatment prior to AZA appears to be more advantageous. This sequential approach warrants further investigation.

## 4. Materials and Methods

### 4.1. Cell Culture Conditions

Two cell lines were used in this study. The MOLM-13 cell line (ACC 554) and the SKM-1 cell line (ACC 547) were derived from the peripheral blood of a 20-year-old and a 76-year-old man with AML following MDS, respectively (both supplied by Leibniz-Institute DSMZ-Deutsche Samsung von Microorganism und Zellkulturen GmbH, Braunschweig, Germany). The sensitive SKM-1 cell lines were adapted to 5-aza-2′-deoxycytidine (DAC, Sigma Aldrich, St. Louis, MO, USA) over a 6-month period, through their repeated passaging in a medium containing DAC, in stepwise increasing concentrations, beginning at 0.1 nM, up to a final concentration of 3 μM. This procedure yielded the DAC-resistant SKM-1/DAC cell variant. The procedure relating to the MOLM-13/DAC cell variant has been described previously [11]. The cell variants were cultivated in RPMI 1640 medium, with L-glutamine, containing 12% fetal bovine serum (both from Gibco, Langley, OK, USA), 100,000 units/L of penicillin and 50 mg/L of streptomycin (both from Sigma Aldrich, St. Louis, MO, USA), at 37 °C, in a humidified atmosphere containing 5% CO_2_.

Human embryonal kidney (HEK) 293 cells (ACC305, supplied by Leibniz-Institute DSMZ-Deutsche Samsung von Microorganism und Zellkulturen GmbH, Braunschweig, Germany) were grown under standard conditions, in DMEM, supplemented with 10% (*v*/*v*) bovine calf serum (Biosera, Cholet, France). The cells were regularly tested for mycoplasma contamination.

### 4.2. Determination of the Number and Viability of the Cells

Sensitive and resistant MOLM-13 and SKM-1 cells were incubated under standard culture conditions with various concentrations of DAC, 5-azacytidine (AZA, Sigma Aldrich, St. Louis, MO, USA), teriflunomide (TFN, Sigma Aldrich, St. Louis, MO, USA), or combinations of these drugs, for 72 h. The cell lines were treated with AZA/DAC/TFN every 24 h. The number and viability of the cells were determined by measuring the plasma membrane integrity of individual cells through changes in the electrical resistance induced by cells passing through the detector in the CASY Model TT Cell Counter (Roche Applied Sciences, Madison, WI, USA), according to the manufacturer’s protocol.

### 4.3. MTS Assay and Determination of IC_50_ Values

Sensitive and resistant MOLM-13 and SKM-1 cells were incubated under standard culture conditions with different concentrations of cytarabine or gemcitabine for 48 h and various concentrations of AZA, DAC, TFN, or their combinations, for 72 h. The cell lines were treated with AZA/DAC/TFN every 24 h, and cytarabine and gemcitabine were added only once. After cultivation, the CellTiter 96^®^ AQueous One Solution cell proliferation assay (MTS assay, Promega, Madison, WI, USA) was used to determine the metabolic activity of the cells, according to the manufacturer’s protocol. The IC_50_ (half-maximal inhibitory concentration) was computed using non-linear regression, according to Equation (1), using the SigmaPlot 8.02 software (Systat Software, Inc., San Jose, CA, USA):(1)$$N=a+A\times exp\left[ln\left(0.5\right)\times {\left(\frac{c}{{IC}_{50}}\right)}^{n}\right]$$
where N in % is the metabolic activity of the cells after drug treatment at concentration c; a + A in % is the metabolic activity of control/untreated cells; A represents the metabolic activity that is suppressed by the respective drug; IC_50_ is the half-maximal inhibitory concentration; and n represents order exponents for cytotoxic effects. The data represent computed values ± standard error, with 30 degrees of freedom.

### 4.4. RNA Isolation and Reverse Transcription

The total RNA was isolated using the TRI Reagent (MRC, Cincinnati, OH, USA), according to the manufacturer’s instructions. Reverse transcription (RT) was performed with 1 μg of RNA, using a RevertAid™ H Minus First-Strand cDNA synthesis kit (Thermo Fisher Scientific, Waltham, MA, USA), according to the manufacturer’s protocol.

### 4.5. Determination of mRNA Gene Expression

The qPCR analysis was performed with a total volume of 10 μL, using the iTaq Universal SYBR Green Supermix (Bio-Rad, Laboratories, Hercules, CA, USA), according to the manufacturer’s protocol, with the primers (final concentration 500 nM) detailed in Table 2. As a template, we used 4 μL of 50X diluted cDNA. For the thermal cycling, a CFX96 Real-Time System C1000 Touch Thermal Cycler (Bio-Rad, Laboratories, Hercules, CA, USA) was used and the thermal cycling conditions were as follows: 95 °C, 30 s and 40 cycles of 95 °C, 5 s and 60 °C, 30 s. All the reactions were performed in triplicate, and the mRNA quantities were normalized to the quantity of *ACTB*. The relative expression of the target genes was calculated using the 2^−ΔΔCt^ method. The data were analyzed with Bio-Rad CFX Manager 3.1 software.

### 4.6. Sequence Analysis

The PCR analysis was performed with a total volume of 20 μL, using a Phusion™ High-Fidelity DNA Polymerase PCR kit (Thermo Fisher Scientific, Waltham, MA, USA), according to the manufacturer’s protocol. The PCR thermal cycling conditions were as follows: initial denaturation (98 °C, 5 min); 30 cycles of denaturation (98 °C, 1 min), annealing (temperature in Table 2, primers marked with *, 30 s) and extension (72 °C, 2 min); followed by a final extension (72 °C, 10 min). The PCR products were separated on a 1.5% agarose gel (Lonza, Rockland, ME, USA), and the gel was visualized with GelRed™ nucleic acid gel stain (Biotium, Fremont, CA, USA), using an Amersham Imager 600 (GE Healthcare Europe GmbH, Pittsburgh, PA, USA). The PCR products were extracted from the gel with a GeneJET Gel Extraction Kit (Thermo Fisher Scientific, Waltham, MA, USA), according to the manufacturer’s protocol. The sequences of the PCR products were determined by Sanger sequencing (Eurofins Genomics Germany GmbH, Ebersberg, Germany).

### 4.7. Detection of Gene Expression at the Protein Level

The proteins were extracted using an RIPA lysis buffer containing 50 mM Tris-Cl (pH 8.0), 1% Triton X-100, 0.5% sodium deoxycholate, 0.1% SDS, 150 mM NaCl, and a protease inhibitor cocktail from Sigma-Aldrich (Saint Louis, MO, USA). The protein concentrations were measured by a Pierce™ BCA Protein Assay kit (Thermo Fisher Scientific, Waltham, MA, USA). The protein samples were separated using sodium dodecyl sulfate–polyacrylamide electrophoresis (SDS–PAGE) in a 12% gel. The proteins were then transferred through electroblotting to a nitrocellulose membrane (GE Healthcare Europe GmbH, Vienna, Austria). The primary antibodies used were UCK1 (HPA050969) and UCK2 (SAB1411384) (both from Sigma Aldrich, St. Louis, MO, USA); DCK (ab96599) and DNMT1 (60B1220.1) (both from Abcam, Cambridge, UK); GAPDH (MAB374) (EMD Millipore Chemicals, Billerica, MA, USA); and γ-H2AX (CST 9718) and α-tubulin (CST 3873) (both from Cell Signaling Technology, Danvers, MA, USA). Goat anti-rabbit (SC-2054) and mouse anti-rabbit antibodies (SC-2357) (both from Santa Cruz Biotechnology, Dallas, TX, USA), and horse anti-mouse (CST 7076) and goat anti-rabbit antibodies (CST 7074) (both from Cell Signaling Technology, Danvers, MA, USA), all conjugated with horseradish peroxidase, served as secondary antibodies. The protein bands were visualized using ECL detection (GE Healthcare Europe GmbH, Vienna, Austria) and an Amersham Imager 600 (GE Healthcare Europe GmbH, Pittsburgh, PA, USA). The protein quantities were established by densitometry, using the U.S. National Institutes of Health ImageJ 1.53t program, and normalized to GAPDH or α-Tubulin.

### 4.8. Preparation of Expression Vectors, Transfection, and Determination of DCKwt and DCKmut Expression

The construct expressing DCKwt (wild-type DCK) was prepared by inserting the DCK coding sequence, amplified from the parental cell line MOLM-13, between the BamHI/XhoI restriction sites of the pcDNA3.1/myc-His A (Thermo Fisher Scientific, Waltham, MA, USA) vector, so that the myc tag was retained at the C-terminus of the expressed protein. An expression vector of mutated DCK (DCKmut) was prepared in the same way, by inserting the coding sequence of DCK amplified from the DAC-resistant subline MOLM-13/DAC. The following primers were used for amplification:

DCK-F BamHI: accaGGATCCGCCACAAGACTAAGGAATGG;

DCK-R XhoI: accaCTCGAGGCAAGATCCAAAGTACTCA.

The plasmids were propagated in chemically competent E. coli DH5α bacteria (Invitrogen, Waltham, MA, USA), extracted using the GeneJET Plasmid Miniprep kit (Thermo Fisher Scientific, Waltham, MA, USA) and validated using Sanger sequencing by Eurofins Genomics, Germany.

The HEK293 cells were transfected by lipofection, using the jetPRIME transfection reagent (Polyplus, Illkirch-Graffenstaden, France), in a ratio of 1:3 (1 μg DNA to 3 μL transfection reagent). In order to compare the protein expression of the mutated and wt protein, 400 ng of plasmid DNA was used together with 40 ng of plasmid DNA expressing eGFP (enhanced green fluorescent protein), which served as a transfection control. The cells were harvested 24 and 48 h after transfection.

For transfection with subsequent inhibition of the proteasome, 600 ng of the respective plasmid DNA was used. After 24 h, the inhibitors, MG132 and bortezomib, were added at a concentration of 10 μM and 20 nM, respectively, for another 24 h.

The proteins were isolated from cells using a standard procedure, and the expression of the wt and mutated protein DCK and eGFP was determined by Western blot analysis (described in: Detection of gene expression at the protein level). For a comparison of the protein expression, 12.5 μg of the total proteins were applied to the gel, and for inhibition of the proteasome, 17.5 μg of the total proteins. An anti-myc antibody (PLA0001, Sigma Aldrich, St. Louis, MO, USA) was used to detect DCKwt/mut, and an anti-GFP antibody (ab290, Abcam, UK) was used to detect eGFP. A goat anti-rabbit antibody (ab205718, Abcam, UK) conjugated with horseradish peroxidase was used as a secondary antibody.

### 4.9. Detection of Cell Death Mode

The cells were incubated with various concentrations of AZA, DAC, TFN, or a combination of these drugs, for 24 or 72 h, under standard culture conditions. The cells were treated with AZA/DAC/TFN every 24 h. After the incubation period, the proportions of apoptotic and necrotic cells were measured using an Annexin V (Roche, Mannheim, Germany)/propidium iodide (Calbiochem, San Diego, CA, USA) assay. The cells were washed with PBS and gently resuspended in binding buffer, containing 0.5 μg/mL FITC-labeled Annexin V. The mixtures were incubated for 15 min at room temperature in the dark. Finally, propidium iodide (final concentration of 0.6 μg/mL) was added to each sample, after which the samples were analyzed by flow cytometry, using an Accuri C6 flow cytometer (BD Bioscience, San Jose, CA, USA).

### 4.10. Global DNA Methylation Status Determination

The cells were incubated with or without 0.5 μM AZA or DAC for 72 h. The cells were treated with AZA/DAC every 24 h. The total DNA was isolated using TRI Reagent (MRC, Cincinnati, OH, USA), according to the manufacturer’s instructions. The global DNA methylation status was determined by a Methylated DNA Quantification Kit (Abcam, Cambridge, UK), according to the manufacturer’s instructions.

### 4.11. Statistical Analysis and Data Processing

Unless otherwise stated, the numerical data are expressed as the mean ± SD of three independent measurements. Statistical significance was assessed using an unpaired Student’s t test. Correlations were determined using Pearson correlation analysis. The SigmaPlot 8.0 software (Systat Software, Inc., San Jose, CA, USA) was used.

## 5. Conclusions

Two DAC-resistant cell variants have been prepared in our laboratory, both showing cross-resistance to AraC and GEM, but not to AZA. Placing these results in the context of previous research by our group and others, we can see that the development of cross-resistance between hypomethylating agents in cell lines is much more common when AZA is the first drug used. Moreover, our DAC-resistant cells seem to be dependent on the pyrimidine synthesis pathway necessary for AZA activation. Therefore, we suggest that further clinical research should focus on the use of AZA after DAC-treatment failure and not the other way around.

Although diminished DCK expression or activity is already recognized as a mechanism of DAC resistance, this study uncovers a previously unreported mutation in the gene, resulting in A180P substitution. In our DAC-resistant sublines, we observed a mutation, A180P, leading to the downregulation of DCK through its degradation by the proteasome. While the results in clinical settings are inconclusive, at the cell line level, the downregulation of DCK appears to play an important role in secondary resistance to DAC. We believe that the expression of DCK in patient samples should also be measured at the protein level, as measuring mRNA expression alone may not be sufficient given the results in regard to cell lines.

The possibility of overcoming the resistance caused by DCK downregulation may lie in the modification of the drug structure into the monophosphate form using protective groups.

## Figures and Tables

**Figure 1 ijms-26-05083-f001:**
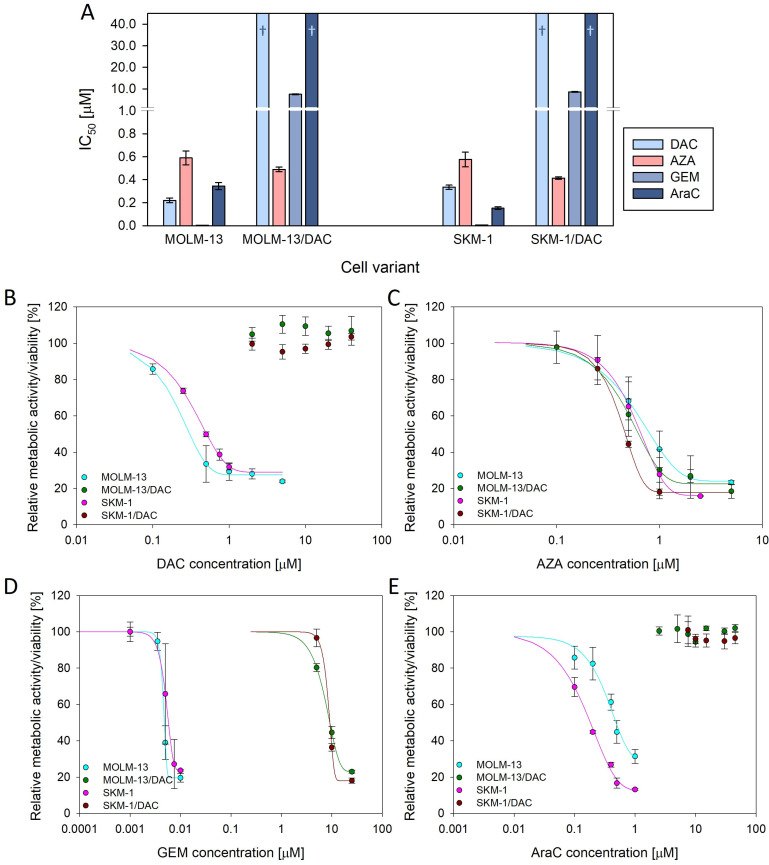
Sensitivity of parental and DAC-resistant cells to 5-aza-2′-deoxycytidine (DAC), 5-azacytidine (AZA), gemcitabine (GEM), and cytarabine (AraC). (**A**) Sensitivity expressed as the half maximal inhibitory concentration (IC_50_). The IC_50_ was calculated using non-linear regression, according to Equation (1), using SigmaPlot. The data represent the calculated value ± standard error, with 30 degrees of freedom. † = Even at a concentration of 40 μM, the drug did not show a half-maximal inhibitory effect on the cells. (**B**–**E**) Dose-response curves for the cell variants after 72 h of cultivation with (**B**) DAC and (**C**) AZA, added at various concentrations every 24 h and after 48 h of cultivation with (**D**) GEM and (**E**) AraC added once.

**Figure 2 ijms-26-05083-f002:**
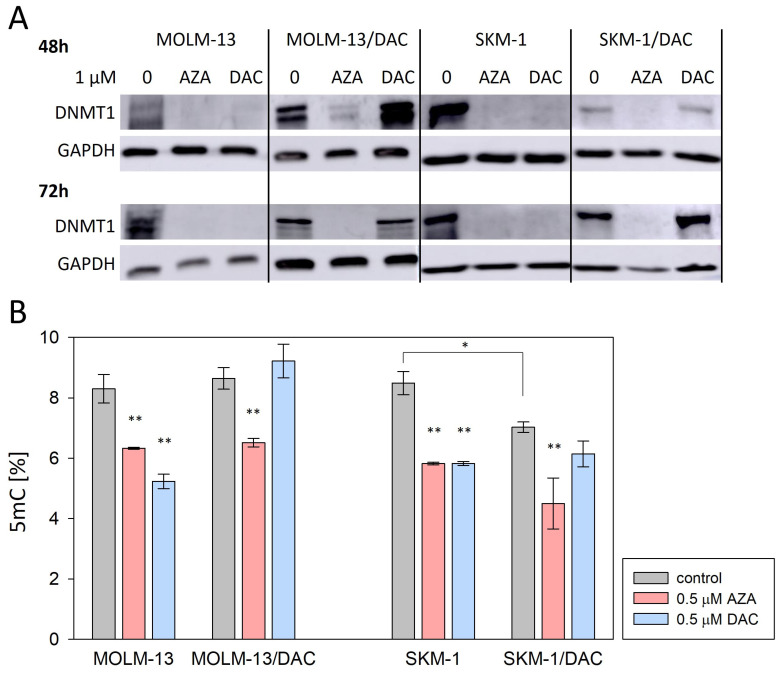
Alterations in DNA methylation induced by HMAs. (**A**) Protein expression of DNMT1 determined by Western blot analysis. GAPDH was used as an internal control. Cells were incubated for 48 or 72 h in the presence or absence of AZA/DAC at 1 μM concentrations. AZA or DAC was added to the cultivation medium every 24 h. The optical densities of the protein bands were quantified by densitometry and are summarized in the bar plots in Appendix A. (**B**) Effect of AZA or DAC on global DNA methylation status, as determined by a methylated DNA quantification kit. Cells were incubated in the presence or absence of AZA/DAC at a concentration of 0.5 μM for 72 h. AZA or DAC was added to the cultivation medium every 24 h. Statistical significance is as follows: * *p* ≤ 0.05; ** *p* ≤ 0.01.

**Figure 3 ijms-26-05083-f003:**
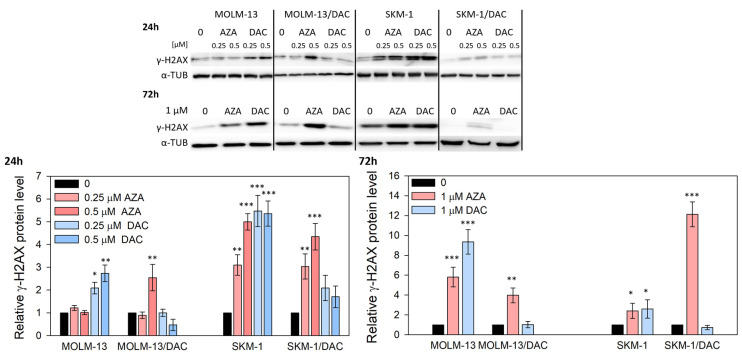
DNA damage induced by HMAs. Relative protein level of phosphorylated H2AX (γ-H2AX) determined by Western blot analysis. The internal control used was α-TUB. Cells were incubated in the presence or absence of AZA/DAC at 0.25 and 0.5 μM concentrations for 24 h and in the presence or absence of AZA/DAC at 1 μM concentration for 72 h. AZA or DAC was added to the cultivation medium every 24 h. The optical densities of the protein bands were quantified by densitometry and are summarized in the bar plots. Statistical significance is as follows: * *p* ≤ 0.05; ** *p* ≤ 0.01; *** *p* ≤ 0.001.

**Figure 4 ijms-26-05083-f004:**
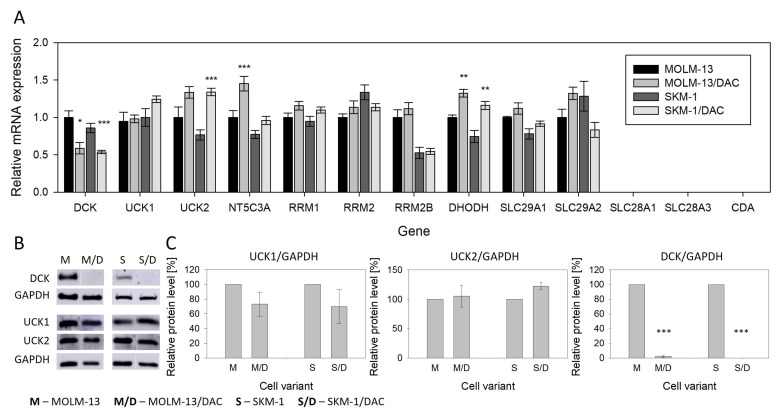
Expression of genes involved in the transport and metabolism of cytidine, deoxycytidine, and their analogs AZA and DAC. (**A**) The mRNA expression of the genes determined by qRT-PCR. *ACTB* was used as an internal control. (**B**) Protein expression of genes UCK1, UCK2, and DCK determined by Western blot analysis. GAPDH was used as an internal control. (**C**) Optical density of the protein bands quantified by densitometry and summarized in the bar plots. The data represent the mean ± SD of three independent measurements. Statistical significance is as follows: * *p* ≤ 0.05; ** *p* ≤ 0.01; *** *p* ≤ 0.001. *CDA*—cytidine deaminase; *NT5C3A*—gene encoding 5′-nucleotidase; *SLC29A1/SLC29A2*—genes encoding human equilibrative nucleoside transporters 1 and 2, respectively; *SLC28A1/SLC28A3*—genes encoding human concentrative nucleoside transporters 1 and 2, respectively; *UCK1/2*—uridine–cytidine kinase 1/2; *DCK*—deoxycytidine kinase; *RRM1/RRM2/RRM2B*—genes encoding ribonucleotide reductase subunits; *DHODH*—dihydroorotate dehydrogenase.

**Figure 5 ijms-26-05083-f005:**
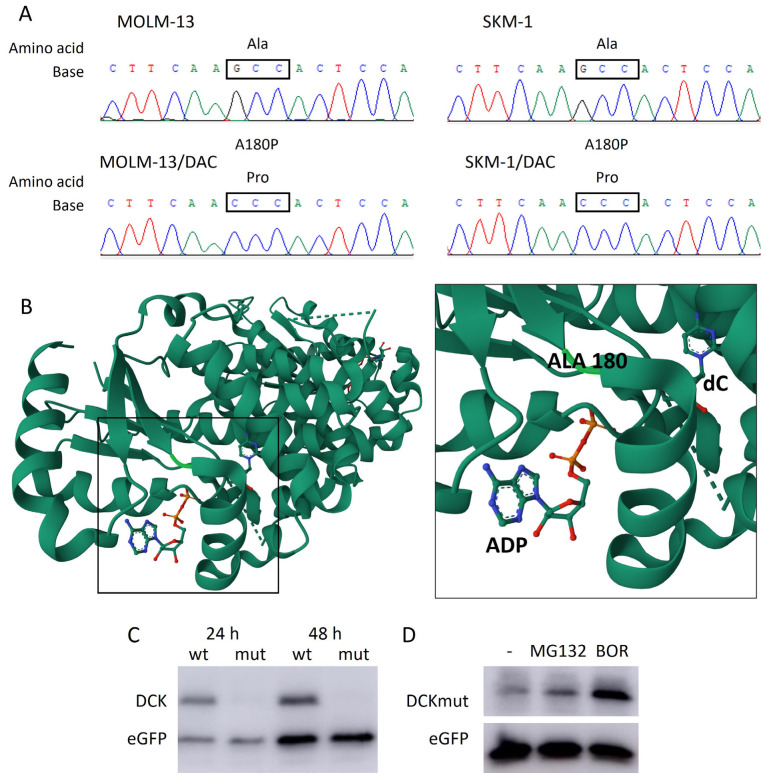
DCK point mutation. (**A**) Results of the RT-PCR product sequence analysis of the DCK coding region in the parental cell lines, MOLM-13 and SKM-1, and the DAC-resistant variants, MOLM-13/DAC and SKM-1/DAC, showing the homozygous mutation site present in the resistant variants leading to the substitution of alanine for proline. (**B**) Crystal structure of human DCK complex with 2′-deoxycytidine (dC) and ADP, with highlighted mutation site (PDB entry: 1P61; [15,16]). (**C**) Protein levels of wild-type (wt) and mutant (mut) DCK in HEK293 cells 24 or 48 h since transfection with expression vectors (400 ng), prepared through the insertion of the amplified DCK coding sequence from the parental cell line, MOLM-13, and the resistant subline, MOLM-13/DAC. (**D**) Protein levels of mutant DCK in HEK293 cells after 24 h of cultivation with the expression vector (600 ng) and subsequent 24 h of cultivation with MG132 (10 μM) or bortezomib (BOR, 20 nM).

**Figure 6 ijms-26-05083-f006:**
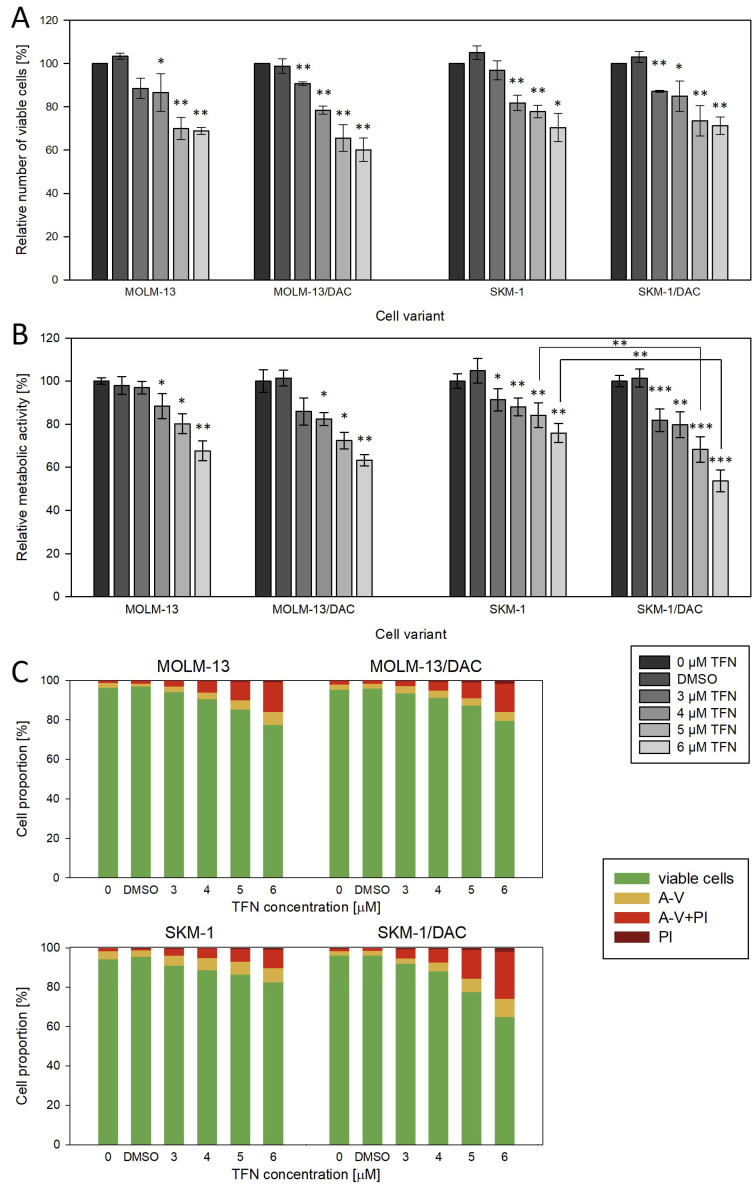
Effect of TFN on DAC-resistant cell variants compared to parental cell lines. Effect was measured after 72 h of cultivation and TFN was added every 24 h. DMSO bars represent the cells treated with the highest concentration of DMSO present during TFN treatment. (**A**) Relative number of viable cells as measured by CASY TT. (**B**) Relative metabolic activity as measured by MTS. (**C**) Proportion of viable (unstained), apoptotic (stained with Annexin-V or both Annexin-V and PI), and necrotic (stained with PI) cells. Representative dot plots are available in Appendix A. Statistical significance is as follows: * *p* ≤ 0.05; ** *p* ≤ 0.01; *** *p* ≤ 0.001.

**Figure 7 ijms-26-05083-f007:**
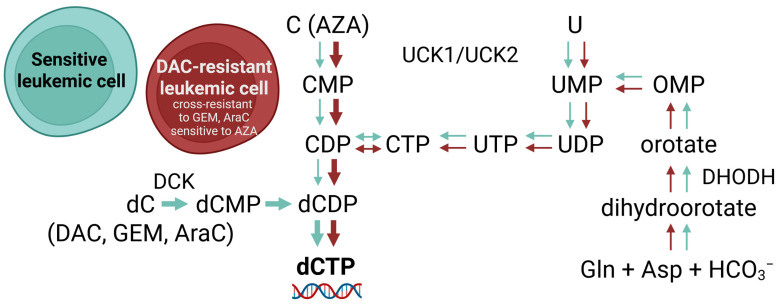
Different dCTP synthesis pathways in sensitive and DAC-resistant cell variants. DAC-resistant cell variants lack the DCK protein, which normally plays a key role in the activation of deoxycytidine. As a result, dCTP must be synthetized by other pathways. The weak inhibitory effect of the DHODH inhibitor, TFN, indicates that these cells are not specifically dependent on the de novo pyrimidine synthesis pathway. However, they seem to synthetize dCTP to a large extent through the salvage pathway from cytidine and uridine, which is supported by the finding concerning the sensitivity of these cells to AZA. Created with BioRender.com. Notes: (d)C—(deoxy)cytidine, U—uridine, Gln—glutamine, Asp—aspartate, HCO_3_^−^—bicarbonate.

**Table 1 ijms-26-05083-t001:** In silico prediction tools used to analyze the A180P mutation in DCK.

In Silico Tool	Predicted Mutation Effect	Score ^a^
MutationTaster [17]	Disease causing (probably deleterious)	Probability value 0.999
PolyPhen-2 [18]	Probably damaging	Score 0.998; sensitivity 0.27; specificity 0.99
SNPs&GO [19]	Disease	Reliability index 4
SIFT [20]	Tolerated	Probability value 0.09

^a^ MutationTaster: value close to 1 indicates high ‘security’ of the prediction; PolyPhen-2: score > 0.90 possibly damaging and score > 0.95 probably damaging; SIFT: value < 0.05 predicted to be deleterious; SNPs&GO reliability index: scoring from 0 (unreliable) to 10 (reliable).

**Table 2 ijms-26-05083-t002:** PCR primers for respective genes.

Gene	Primer Sequences (5′-3′)	T_A_ (°C)	PCR Product Size (bp)
*ACTB*	GACGACATGGAGAAAATCTG ATGATCTGGGTCATCTTCTC	60	131
*CDA*	GCAACATAGAAAATGCCTGCT TAGCAATTGCCCTGAAATCC	60	102
*DCK*	GTCTCAGAAAAATGGTGGGAATG ACAGGTTTCTCTGCATCTTTGAG	60	150
*UCK1*	CGTGTGTGAGAAGATCATGG TGGTCAAAATTGTACTGTCCTTT	60	150
*UCK2*	GACATCAGCGAGAGAGGCAG TCTTGCGTGAAGGGGTGTAG	60	187
*NT5C3A*	GTCAAGCCTGCCTTTGAGGA TTCCTCAAGGCACCATCATGT	60	198
*RRM1*	TGGAATTGGGGTACAAGGTC GAGAGCCCTCATAGGTTTCG	60	176
*RRM2*	TTTACACTGTGATTTTGCTTGC TGTTCTATCCGAACAGCATTG	60	102
*RRM2B*	ATAAACAGGCACAGGCTTCC GAACCTGCACCTCCTGACTAA	60	186
*SLC29A1*	CATGCTGCCCCTGCTGTTAT CTGCACCTTCACCAGGATGG	60	136
*SLC29A2*	TCCTTCCTGTACCAGTGCGT ATGTCCACCTTGACCAGCG	60	104
*SLC28A1*	AGGTTCTGCCCATCATTGTC CAAGTAGGGCCGGATCAGTA	60	197
*SLC28A3*	GACTCACATCCATGGCTCCT TTCCAGGGAAAGTGGAGTTG	60	183
*DHODH*	CTGAACACCTGATGCCGACT AGTCTTGAAATCTGGCCCGT	60	111
*DCK* * ^a^ *	CAGGATCTGGCTTAGCGG CATTTGGCTGCCTGTAGTCT	63	914
*UCK1* * ^a^ *	AGATGGCTTCGGCGGGA AGTCCCTGAACACACATGCC	65	890
*UCK2* * ^a^ *	AACCATGGCCGGGGACAG GATGAGCAGTGCCTCCTGAC	65	858

*^a^*—primers used for sequence analysis.

## Data Availability

The original contributions presented in this study are included in the article/Appendix A. Further inquiries can be directed to the corresponding authors.

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
