# Peer review of "Acquired Resistance to Decitabine Associated with the Deoxycytidine Kinase A180P Mutation: Implications for the Order of Hypomethylating Agents in Myeloid Malignancies Treatment"

_ijms, 2025, doi:10.3390/ijms26115083_

Round 1
Reviewer 1 Report
Comments and Suggestions for Authors
Simonicova et al has submitted a manuscript entitled “Acquired Resistance to Decitabine Associated with the Deox-2 ycytidine Kinase A180P Mutation: Implication for the Order of 3 Hypomethylating Agents in Myeloid Malignancies Treatment” for publication. This study describes AML cells that show cross-resistance to cytarabine and gemcitabine, but to 5-azacytidine. A key molecular feature of resistant cells was a point mutation (A180P) in the deoxycytidine kinase (DCK) gene that leads to degradation of the protein. These suggests that 5-azacytidine may represent a treatment option for patients with acquired DAC resistance.03 Even though this is an interesting mechanism of resistance, there are key experiments missing that lowers the impact of this manuscript. Listed below are my specific concerns.
- Over expression of the mutant protein in a non-AML cell lines are not enough. Wild type DCK needs to be expressed in the resistant AML cell lines and determine whether DAC sensitivity restored.
- The proteosome inhibitors suggest that DCK mutant is being actively degraded. This is however only in HEK 293 cells. This needs to be repeated in the AML cell lines.
- Conversely, DCK needs to be knocked down and the ability of these cells to be sensitive to DAC treatment must be evaluated.
- The AZA treatments need to be evaluated in more detail. Is the cell death apoptosis and is it affecting mitochondria or ROS.
- Need to show flow cytometry data for TFN treatment in addition to bar graphs.
- It is unclear whether DCK gene mutated is detected in AML relapse?
- Finally, analysis of DNA damage in the resistance cells needs be evaluated.
Author Response
Dear Reviewer,
Thank you for your valuable and constructive feedback on our manuscript. We truly appreciate the time and effort you devoted to reviewing our work. Please find our detailed responses to your comments enclosed.
Simonicova et al has submitted a manuscript entitled “Acquired Resistance to Decitabine Associated with the Deox-2 ycytidine Kinase A180P Mutation: Implication for the Order of 3 Hypomethylating Agents in Myeloid Malignancies Treatment” for publication. This study describes AML cells that show cross-resistance to cytarabine and gemcitabine, but to 5-azacytidine. A key molecular feature of resistant cells was a point mutation (A180P) in the deoxycytidine kinase (DCK) gene that leads to degradation of the protein. These suggests that 5-azacytidine may represent a treatment option for patients with acquired DAC resistance.03 Even though this is an interesting mechanism of resistance, there are key experiments missing that lowers the impact of this manuscript. Listed below are my specific concerns.
1. Over expression of the mutant protein in a non-AML cell lines are not enough. Wild type DCK needs to be expressed in the resistant AML cell lines and determine whether DAC sensitivity restored.
Ans: We thank the reviewer for this thoughtful suggestion. We agree that re-expression of wild-type DCK in resistant AML cell lines would provide further insight into the functional consequences of the A180P mutation. However, MOLM-13 and SKM-1 cells—like many other suspension cell lines—are technically difficult to transfect efficiently, and introducing wild-type DCK in these models would pose several challenges. While transfection of AML cell lines is possible, it involves challenges such as low efficiency and high cytotoxicity. These factors can complicate experiments, especially those requiring prolonged drug treatments like with DAC, which necessitates repeated dosing over several days. Such an experiment would require extensive optimalisation, with no guaranteed outcome. Moreover, the association between DCK downregulation and DAC resistance has already been demonstrated in the literature [1–3] and we observed the A180P mutation independently in two different resistant cell variants. Our use of HEK293 cells, chosen for their ease of transfection, was not intendent to model resistance, but rather to demonstrate the effect of the novel A180P mutation on DCK protein stability.
2. The proteosome inhibitors suggest that DCK mutant is being actively degraded. This is however only in HEK 293 cells. This needs to be repeated in the AML cell lines.
Ans: We thank the reviewer for this important comment. We agree that confirming DCK degradation in AML cell lines would strengthen our findings. We did attempt these experiments; however, our AML cell lines are highly sensitive to proteasome inhibitors and undergo apoptosis even at low concentrations of bortezomib [4], which makes it challenging to effectively inhibit the proteasome without inducing significant cell death. In contrast, HEK293 cells tolerate these concentrations and allowed us to observe the accumulation of the mutated DCK protein. Furthermore, it is important to note that in HEK293 cells we examined overexpressed DCK, which is present at much higher levels than the endogenous DCK in AML cell lines. This higher expression facilitates detection and monitoring of protein stability and degradation. In AML cells, the lower endogenous DCK levels further complicate analysis, especially under conditions of proteotoxic stress and rapid cell death. Despite these limitations, our results in HEK293 provide a useful model for understanding the degradation dynamics of DCK mutant. We added a clarifying statement into discussion section, lines 346-350.
3. Conversely, DCK needs to be knocked down and the ability of these cells to be sensitive to DAC treatment must be evaluated.
Ans: We thank the reviewer for the suggestion. DCK knockdown and its effect on DAC sensitivity have been studied in the literature, with several publications demonstrating that reduced DCK expression is associated with DAC resistance [1–3]. Given this established understanding, we did not feel that repeating this experiment would add significant new information to our study. Instead, we focused on characterizing the mutations in the DCK gene in DAC-resistant cell lines, which may provide novel insights into the mechanisms of drug resistance.
4. The AZA treatments need to be evaluated in more detail. Is the cell death apoptosis and is it affecting mitochondria or ROS.
Ans: We thank the reviewer for the suggestion. The mechanism of action of HMAs was, however, not the focus of this study. In our previous work, we demonstrated that both AZA and DAC induce apoptosis and reduce mitochondrial membrane potential in the parental MOLM-13 cell line, and that AZA induces the same effects in the resistant MOLM-13/DAC cell line [4].
5. Need to show flow cytometry data for TFN treatment in addition to bar graphs.
Ans: We thank the reviewer to point this out. Flow cytometry data were added to supplementary file as Figure S2 and Figure S3.
6. It is unclear whether DCK gene mutated is detected in AML relapse?
Ans: We thank the reviewer for the comment. To our knowledge, there are currently no published studies that have specifically investigated DCK mutations in AML patients who relapsed after DAC treatment. One relevant study by Qin et al. (2011) analyzed 16 MDS patients who relapsed following DAC therapy and did not detect DCK mutations. However, the clinical context of those cases is not fully described — including how relapse was defined, how many treatment cycles patients received prior to relapse, and which 16 of the 30 relapsed patients summarized in the table were selected for sequencing. Therefore, while this study provides some insight, its conclusions are limited.
As we mentioned in the Discussion section, published studies to date have mainly focused on DCK mRNA expression in relapsed patients, while protein expression and enzymatic activity of DCK have not been investigated. We now added information about DCK mutation status assessment (lines 365-371).
Unfortunately, as stated in the Discussion section, we were not able to assess the mutation status in relapsed patients, as we currently have no access to clinical samples from patients treated with DAC (DAC is not reimbursed by health insurance in Slovak and Czech Republic, and therefore, unlike AZA, it is not commonly used in clinical practice). One of the motivations behind this study is to raise awareness of this issue and encourage other researchers who have access to such patient material to further investigate the potential role of DCK mutations in patients who relapse following DAC treatment.
7. Finally, analysis of DNA damage in the resistance cells needs be evaluated.
Ans: We thank the reviewer for suggesting further analysis of DNA damage. In our study, we used γ-H2AX as a sensitive and widely accepted early marker of DNA double-strand breaks to assess the induction of DNA damage by DAC and AZA. As shown in Figure 2B (now Figure 3), phosphorylation of H2AX (γ-H2AX) was clearly increased in sensitive cells by both AZA and DAC and in resistant cells treated with AZA, but remained unchanged in the resistant variants treated with DAC, indicating the absence of a DNA damage in these cells caused by DAC.
Given that H2AX phosphorylation represents an upstream event in the DNA damage repair pathway, the lack of its induction suggests that downstream signaling would also likely not occur. Furthermore, the use of γ-H2AX is well-established in studies of this kind. Therefore, we believe that γ-H2AX is sufficient to support our conclusion that DAC does not induce DNA damage in resistant cells while AZA does.
In this paper we only wanted highlight that in DAC-resistant variants AZA had similar effects as in parental cell variants, while DAC did not cause these effects. The mechanisms of DNA damage caused by HMAs would be an interesting area to study, and we hope to address this problem in the future, however, it is beyond the scope of this particular paper.
References:
- Zhang, P.; Zhang, Z.; Wang, Y.; Du, W.; Song, X.; Lai, W.; Wang, H.; Zhu, B.; Xiong, J. A CRISPR-Cas9 Screen Reveals Genetic Determinants of the Cellular Response to Decitabine. EMBO Rep. 2025, 26, 1528, doi:10.1038/s44319-025-00385-w.
- Dahn, M.L.; Cruickshank, B.M.; Jackson, A.J.; Dean, C.; Holloway, R.W.; Hall, S.R.; Coyle, K.M.; Maillet, H.; Waisman, D.M.; Goralski, K.B.; et al. Decitabine Response in Breast Cancer Requires Efficient Drug Processing and Is Not Limited by Multidrug Resistance. Mol. Cancer Ther. 2020, 19, 1110–1122, doi:10.1158/1535-7163.MCT-19-0745.
- Gruber, E.; Franich, R.L.; Shortt, J.; Johnstone, R.W.; Kats, L.M. Distinct and Overlapping Mechanisms of Resistance to Azacytidine and Guadecitabine in Acute Myeloid Leukemia. Leukemia 2020, 34, 3388–3392, doi:10.1038/s41375-020-0973-z.
- Janotka, Ľ.; Messingerová, L.; Šimoničová, K.; Kavcová, H.; Elefantová, K.; Sulová, Z.; Breier, A. Changes in Apoptotic Pathways in MOLM‐13 Cell Lines after Induction of Resistance to Hypomethylating Agents. Int. J. Mol. Sci. 2021, 22, 1–25, doi:10.3390/ijms22042076.
Reviewer 2 Report
Comments and Suggestions for Authors
In this manuscript titled "Acquired resistance to decitabine associated with the Deoxycytidine kinase A180P mutation: implication for the order of Hypometylation agents in myeloid malignancies treatment", a team led by Simonicova and Messingerova investigate the mechanism of HMAs resistance in AML cell lines.
The author made several interesting points here based on the previously published study. There were couple of significant weaknesses that preclude it can be accepted on the current version.
- From line 103 to line 116. The author needed to re-organize the description to refer to the order of the display of figures.
- The time points of panel A and panel B in Figure 2 aren't consistent. Is there any special reason or not? The author needs to explain or re-make the figure.
- In panel A of Figure 2, the band of DNMT1 on 72h in MOLM-13 is too vague. In addition, what about other methylation makers' expressions? such as DNMT3A, EZH2, MBD1, MBD4, or TET1. Besides, in panel B of Figure 2, the phospho-H2AX should be measured, not H2AX. Besides, what about the expression of other DNA damage markers? The DNA damage antibody sampler kit (#9947, cell signaling technology) may be considered for this experiment.
- The Bar graph in Figure 2 may not necessarily be shown in the main text since the trend is clear. It can be moved to the supplementary.
- Based on the general rules of the articles, for the western blot, the GAPDH or TUB should be listed after your target gene. The author needs to re-organize all the western blot in this manuscript.
- In Figure 5B, the author needs to explain how to measure the cell metabolism by MTS? the MTS assay, supposed to measure the cell viability, can not used for the metabolism. Besides, the methods also need to be added to this section for metabolism detection.
Author Response
Dear Reviewer,
Thank you for your valuable and constructive feedback on our manuscript. We truly appreciate the time and effort you devoted to reviewing our work. Please find our detailed responses to your comments enclosed.
In this manuscript titled "Acquired resistance to decitabine associated with the Deoxycytidine kinase A180P mutation: implication for the order of Hypometylation agents in myeloid malignancies treatment", a team led by Simonicova and Messingerova investigate the mechanism of HMAs resistance in AML cell lines.
The author made several interesting points here based on the previously published study. There were couple of significant weaknesses that preclude it can be accepted on the current version.
- From line 103 to line 116. The author needed to re-organize the description to refer to the order of the display of figures.
Ans: We thank the reviewer for pointing this out. We have reorganized the entire Chapter 2.2, incorporating suggestions from this and other reviewer comments.
2. The time points of panel A and panel B in Figure 2 aren't consistent. Is there any special reason or not? The author needs to explain or re-make the figure.
Ans: We thank the reviewer for this helpful comment. The time points used in the original Figure 2 were chosen based on the distinct kinetics of the biological processes being analyzed. DNMT1 protein expression (originally Panel A) was assessed at 48 and 72 hours, as changes in DNA methylation and protein levels typically occur over a longer time frame following drug treatment. In contrast, γ-H2AX phosphorylation (originally Panel B) is an early marker of DNA damage, and thus was measured at 24 and 72 hours to capture both early and sustained responses to hypomethylating agents.
To improve clarity and avoid confusion caused by the differing time points, we have now separated the two panels into individual figures—Figure 2 (DNMT1 expression and global DNA methylation) and Figure 3 (γ-H2AX levels). We believe this change enhances the readability and logical flow of the data presentation.
3. In panel A of Figure 2, the band of DNMT1 on 72h in MOLM-13 is too vague. In addition, what about other methylation makers' expressions? such as DNMT3A, EZH2, MBD1, MBD4, or TET1. Besides, in panel B of Figure 2, the phospho-H2AX should be measured, not H2AX. Besides, what about the expression of other DNA damage markers? The DNA damage antibody sampler kit (#9947, cell signaling technology) may be considered for this experiment.
Ans: We thank the reviewer for pointing out the poor quality of the DNMT1 band in MOLM-13 at the 72-hour time point. We have adjusted the image to improve its clarity and hope that the changes are satisfactory.
We appreciate the reviewer’s suggestion to investigate additional regulators of the epigenetic machinery. While proteins such as DNMT3A, EZH2, and MBD family members are important for DNA methylation dynamics, our study aimed to assess whether the drug modulates global DNA methylation in resistant cells. Our current data — showing both the lack of DNM1 downregulation and unchanged global DNA methylation in resistant cells — strongly suggest that the drug fails to induce epigenetic reprogramming in this context. While mechanistic studies involving additional epigenetic regulators would be valuable, they fall outside the scope of the current investigation.
Regarding the comment on H2AX, we would like to clarify that in Figure 2B (now Figure 3) we indeed measured the phosphorylated form of H2AX (commonly referred to as γ-H2AX in the literature), which is a well-established marker of DNA damage. This was assessed using an antibody specific to H2AX phosphorylated at Ser139. We have revised the figure legend slightly to ensure this point is communicated more clearly to avoid any potential confusion.
We thank the reviewer for suggesting the inclusion of additional DNA damage response markers such as phosphorylated ATM, BRCA1, Chk1, Chk2, and p53. In our study, we used γ-H2AX as a sensitive and widely accepted early marker of DNA double-strand breaks to assess the induction of DNA damage by DAC and AZA. As shown in Figure 3 (previously Figure 2B), γ-H2AX was clearly induced in sensitive cells by both AZA and DAC and in resistant cells treated with AZA, but remained unchanged in the resistant variants treated with DAC, indicating the absence of a DNA damage in these cells after DAC treatment.
Given that γ-H2AX phosphorylation represents an upstream event in the DDR pathway, the lack of its induction suggests that downstream signaling through ATM and related factors would also likely not occur. Therefore, we believe that γ-H2AX is sufficient to support our conclusion that the drug does not induce DNA damage in resistant cells. In this paper we only wanted highlight that in DAC-resistant variants AZA had similar effects as in parental cell variants, while DAC did not cause these effects. The mechanisms of DNA damage caused by HMAs would be an interesting area to study, and we hope to address this problem in the future, however, it is beyond the scope of this particular paper.
4. The Bar graph in Figure 2 may not necessarily be shown in the main text since the trend is clear. It can be moved to the supplementary.
Ans: We thank the reviewer for the suggestion to move the bar graph in Figure 2 to the supplementary materials. We agree that in some cases, such graphs may be redundant when the trend is clear from the blots. However, based on our previous experience, many reviewers require quantification of western blot data, even when the visual trend is obvious. We decided to move the bar graph from Figure 2A to the supplementary data, as suggested. However, we have retained the bar graph corresponding to Figure 3 (previously Figure 2B), as we believe it might provide additional clarity in interpreting the results.
5. Based on the general rules of the articles, for the western blot, the GAPDH or TUB should be listed after your target gene. The author needs to re-organize all the western blot in this manuscript.
Ans: We thank the reviewer for the helpful suggestion. We have reorganized all Western blot figures in the manuscript so that the loading control (GAPDH or TUB) is shown after the target gene in accordance with the general formatting conventions for Western blot presentations. During this process, we also noticed that in Figure 3, now Figure 4, the representative image was taken from a different replicate than the one used in the uncropped membranes. We have now replaced it with the correct image. This change, however, does not affect the overall results.
6. In Figure 5B, the author needs to explain how to measure the cell metabolism by MTS? the MTS assay, supposed to measure the cell viability, can not used for the metabolism. Besides, the methods also need to be added to this section for metabolism detection.
Ans: We thank the reviewer for the comment. We respectfully clarify that the MTS assay, while commonly used as a cell viability assay, fundamentally measures the metabolic activity of cells. The assay relies on the bioreduction of the MTS tetrazolium compound into a soluble formazan product by NAD(P)H-dependent dehydrogenase enzymes, which are active in metabolically functioning cells. Thus, the assay reflects mitochondrial activity and overall metabolic function, making it a valid method to assess cell metabolism, particularly under conditions where mitochondrial function is expected to change. Numerous studies and assay protocols support the use of MTS as a proxy for cellular metabolic activity [1,2]. In our laboratory, we have also observed that certain compounds at specific concentrations increase MTS signal without affecting cell number, suggesting a direct effect on cellular metabolism. This relationship between MTS signal and metabolic activity has been further discussed in papers of our colleagues [3,4].
References:
- Sieuwerts, A.M.; Klijn, J.G.M.; Peters, H.A.; Foekens, J.A. The MTT Tetrazolium Salt Assay Scrutinized: How to Use This Assay Reliably to Measure Metabolic Activity of Cell Cultures in Vitro for the Assessment of Growth Characteristics, IC50-Values and Cell Survival. Clin. Chem. Lab. Med. 1995, 33, 813–824, doi:10.1515/cclm.1995.33.11.813.
- Chan, G.K.Y.; Kleinheinz, T.L.; Peterson, D.; Moffat, J.G. A Simple High-Content Cell Cycle Assay Reveals Frequent Discrepancies between Cell Number and ATP and MTS Proliferation Assays. PLoS One 2013, 8, e63583, doi:10.1371/journal.pone.0063583.
- Kontar, S.; Imrichova, D.; Bertova, A.; MacKova, K.; Poturnayova, A.; Sulova, Z.; Breier, A. Cell Death Effects Induced by Sulforaphane and Allyl Isothiocyanate on P-Glycoprotein Positive and Negative Variants in L1210 Cells. Molecules 2020, 25, 2093, doi:10.3390/molecules25092093.
- Bertova, A.; Kontar, S.; Ksinanova, M.; Vergara, A.Y.; Sulova, Z.; Breier, A.; Imrichova, D. Sulforaphane and Benzyl Isothiocyanate Suppress Cell Proliferation and Trigger Cell Cycle Arrest, Autophagy, and Apoptosis in Human AML Cell Line. Int. J. Mol. Sci. 2024, 25, 13511, doi:10.3390/ijms252413511.
Reviewer 3 Report
Comments and Suggestions for Authors
In this paper, the authors correlates the onset of Deox- 2ycytidine Kinase mutation and the resistance to Decitabine.
The paper is interesting and prompt new studies on HMA order of administration. Following some observations:
The abstract only contains the results of the study. A more structured ones (including background, methods, results and conclusions section would be more readable).
“Myelodysplastic syndromes (MDS), or newly myelodysplastic neoplasms..” I did not ubderstand what the authors refer to…
Mind cite the figures and panels in order of appearance.
Chapter 2.2 the authors should better explain, before describing the experiments, the choice of using histone H2AX phosphorylation as biomarkers of HMA incorporation into DNA.
Chapter 2.3 when describing the genes tested, the authors refer to “HMA-activating and -deactivating enzyme”; I think that a division between activating and deactivating would be useful for better clarity and help interpreting the results.
Author Response
Dear Reviewer,
Thank you for your valuable and constructive feedback on our manuscript. We truly appreciate the time and effort you devoted to reviewing our work. Please find our detailed responses to your comments enclosed.
In this paper, the authors correlates the onset of Deox- 2ycytidine Kinase mutation and the resistance to Decitabine. The paper is interesting and prompt new studies on HMA order of administration. Following some observations:
- The abstract only contains the results of the study. A more structured ones (including background, methods, results and conclusions section would be more readable).
Ans: We thank the reviewer for this valuable suggestion. We have revised the abstract to follow a more structured format as much as possible within the 200-word limit. However, we did not include section headings, as the Instructions for Authors specify that the abstract should follow the style of a structured abstract, but without headings.
- “Myelodysplastic syndromes (MDS), or newly myelodysplastic neoplasms..” I did not ubderstand what the authors refer to…
Ans: We thank the reviewer for pointing out that this might not be clear to readers. We refer to “The 5th edition of the World Health Organization Classification of Haematolymphoid Tumours: Myeloid and Histiocytic/ Dendritic Neoplasms” where the new terminology was introduced for MDS:
“The classification introduces the term myelodysplastic neoplasms (abbreviated MDS) to replace myelodysplastic syndromes, underscoring their neoplastic nature and harmonizing terminology with MPN.” [1]
We have revised the manuscript to improve clarity for readers.
- Mind cite the figures and panels in order of appearance.
Ans: We thank the reviewer for pointing this out. We have reorganized the description in text to align with the order of the figures displayed, ensuring better clarity and flow.
- Chapter 2.2 the authors should better explain, before describing the experiments, the choice of using histone H2AX phosphorylation as biomarkers of HMA incorporation into DNA.
Ans: We thank the reviewer for this valuable comment. In response, we have revised the entire Chapter 2.2, incorporating suggestions from this and other reviewer comments.
- Chapter 2.3 when describing the genes tested, the authors refer to “HMA-activating and -deactivating enzyme”; I think that a division between activating and deactivating would be useful for better clarity and help interpreting the results.
Ans: We thank the reviewer for this helpful suggestion. In the revised manuscript, we have separated the enzymes into distinct functional categories: HMA activation and HMA deactivation and we added titles for other categories as well. We have also included some additional explanation for clarification.
Refences:
- Khoury, J.D.; Solary, E.; Abla, O.; Akkari, Y.; Alaggio, R.; Apperley, J.F.; Bejar, R.; Berti, E.; Busque, L.; Chan, J.K.C.; et al. The 5th Edition of the World Health Organization Classification of Haematolymphoid Tumours: Myeloid and Histiocytic/Dendritic Neoplasms. Leukemia 2022, 36, 1703–1719, doi:10.1038/s41375-022-01613-1.
Round 2
Reviewer 1 Report
Comments and Suggestions for Authors
The author failed to address my concerns. The transient transfection of AML cell lines is a challenge but stable transfections are possible. Without the suggested experiments the paper is not acceptable.
Author Response
Dear Reviewer,
We sincerely thank you for your detailed and thoughtful feedback on our manuscript. We acknowledge your point regarding the importance of demonstrating whether re-expression of wild-type DCK in the DAC-resistant AML cell lines would restore DAC sensitivity.
Following the editor’s guidance, we have now explicitly acknowledged the absence of these re-expression experiments as a limitation of our study in the revised discussion section (lines 337–345), where we also explain the technical barriers and point to relevant prior studies demonstrating the link between DCK loss and DAC resistance. We hope that this transparent acknowledgment sufficiently addresses your concern.
We truly appreciate your thoughtful comments, which helped improve the clarity and rigor of our manuscript.
Reviewer 2 Report
Comments and Suggestions for Authors
The author addressed all the comments, and the manuscript has improved significantly.
Author Response
Dear Reviewer,
We thank you very much for your careful reading of our manuscript and for your constructive feedback. We are pleased that our revisions and clarifications addressed your comments.
Thank you again for your valuable input and time.